# Exercise and incidence of myocardial infarction, stroke, hypertension, type 2 diabetes and site-specific cancers: prospective cohort study of 257854 adults in South Korea

Youngwon Kim,[1,2] Stephen Sharp,[2] Semi Hwang,[1,3] Sun Ha Jee[4]

[1]School of Public Health, The University of Hong Kong Li Ka Shing Faculty of Medicine, Hong Kong, Hong Kong
[2]MRC Epidemiology Unit, University of Cambridge School of Clinical Medicine, Cambridge, UK
[3]Department of Epidemiology and Health Promotion, Graduate School of Public Health, Yonsei University, Seoul, South Korea
[4]Epidemiology, Graduate School of Public Health, Yonsei University, Seoul, South Korea

**Correspondence to**
Dr Sun Ha Jee; jsunha@yuhs.ac

## ABSTRACT

**Objective** The objective of this study was to examine the longitudinal associations of exercise frequency with the incidence of myocardial infarction, stroke, hypertension, type 2 diabetes and 10 different cancer outcomes.

**Design** A prospective cohort study.

**Setting** Physical examination data linked with the entire South Korean population's health insurance system: from 2002 to 2015.

**Participants** 257854 South Korean adults who provided up to 7 repeat measures of exercise (defined as exercises causing sweat) and confounders.

**Primary outcome measures** Each disease incidence was defined using both fatal and non-fatal health records (a median follow-up period of 13 years).

**Results** Compared with no exercise category, the middle categories of exercise frequency (3–4 or 5–6 times/week) showed the lowest risk of myocardial infarction (HR 0.79; 95% CI 0.70 to 0.90), stroke (HR 0.80; 95% CI 0.73 to 0.89), hypertension (HR 0.86; 95% CI 0.85 to 0.88), type 2 diabetes (HR 0.87; 95% CI 0.84 to 0.89), stomach (HR 0.87; 95% CI 0.79 to 0.96), lung (HR 0.80; 95% CI 0.71 to 0.91), liver (HR 0.85; 95% CI 0.75 to 0.98) and head and neck cancers (HR 0.76; 95% CI 0.63 to 0.93; for 1–2 times/week), exhibiting J-shaped associations. There was, in general, little evidence of effect modification by body mass index, smoking, alcohol consumption, family history of disease and sex in these associations.

**Conclusions** Moderate levels of sweat-inducing exercise showed the lowest risk of myocardial infarction, stroke, hypertension, type 2 diabetes, stomach, lung, liver and head and neck cancers. Public health and lifestyle interventions should, therefore, promote moderate levels of sweat-causing exercise as a behavioural prevention strategy for non-communicable diseases in a wider population of East Asians.

## INTRODUCTION

Prevention and control of non-communicable diseases are a contemporary global public health priority. At present, 40million deaths per year, which accounts for nearly 70% of total deaths globally, are attributable

### Strengths and limitations of this study

► This study is the first to investigate the longitudinal associations of exercise with various cardiovascular disease and cancer incidence using a large-scale cohort dataset of South Korean adults (n=257854) who provided up to seven repeated measures of exercise and all confounders in order to minimise the risk of regression dilution.
► A limitation is that no strong causal inference can be drawn about the exercise–incident disease relationships.
► Findings of this study may not be generalisable to adult populations of other ethnic origins.

to non-communicable diseases.[1 2] Moreover, the number of deaths due to non-communicable diseases, such as cardiovascular disease (CVD),[3] hypertension,[4] diabetes[5] and cancer,[6] has increased dramatically over the past few decades, although age-standardised CVD and cancer rates as well as systolic blood pressure levels[7] have declined.[8 9] However, trends in these disease traits have varied across different populations, particularly with less favourable changes observed in East Asian populations compared with Western populations. For example, the prevalence of diabetes[10] has increased more rapidly, while the age-standardised prevalence of CVD[3] and systolic blood pressure levels[7] have fallen less steeply in East Asians in comparison with Westerners.

In addition, adults in East Asia tend to have higher prevalence of physical inactivity,[11] which is one of the four target behaviours (including unhealthy diet, tobacco use and harmful use of alcohol) that have been set as the global focus to reduce the risk of non-communicable diseases.[12] The beneficial impacts of increased physical activity on

various non-communicable outcomes have been demonstrated by numerous previous investigations. However, the majority of previous research has been predicated on evidence from Western populations, thereby limiting its application to other populations including East Asians. As such, little is currently known about levels of physical activity including exercise in relation to non-communicable diseases in East Asian populations as compared with Western populations.[13] Another critical gap in the existing literature is the use of data measured only at a single point in time (ie, baseline), in which case physical activity or exercise levels are assumed to remain constant over time. This methodology, therefore, precludes the fact that individuals' physical activity or exercise levels change with time, and hence may increase the potential for regression dilution.[14] Furthermore, it is well known that temporal changes occur in other traditional behavioural and metabolic risk factors for non-communicable diseases, such as adiposity levels,[15 16] smoking,[17] glucose levels[18] and total cholesterol levels,[19] exhibiting different patterns of changes between East Asian and Western populations. Nevertheless, no previous research of East Asians or Westerners took into account changes in these risk markers in understanding the relationships between physical activity and non-communicable diseases. Moreover, the dose–response relationship between physical activity and various non-communicable disease outcomes has remained unclear in East Asians. Therefore, the purpose of this research was to explore the dose–response relationships between exercise frequency and various types of incident non-communicable diseases, such as myocardial infarction, stroke, hypertension, type 2 diabetes and site-specific cancers, using a large-scale prospective cohort of South Korean adults with multiple repeated measures of exercise frequency and other risk markers.

## METHODS

### Study design and participants

This study is based on data from the National Health Insurance Service—Health Screening (NHIS-HEALS) cohort dataset,[20] which is a nationally representative random sample (stratified by 2 groups of sex (males and females), 18 groups of age ranges (less than 1 year, 1–4 years, every 5 years between 5 and 79 years, and more than 80 years), 3 groups of employment status (insured employees, self-employed individuals and medical aid beneficiaries) and 41 groups of income levels (upper 20% for insured employees, lower 20% for insured self-employed individuals and the lowest level for medical aid beneficiaries))[21] of over 500 000 South Korean adults aged 40–79 years between 2002 and 2003 made available by the NHIS. The NHIS is a single health insurance system in South Korea, which manages and maintains information on the entire South Korean population's healthcare utilisation; it is mandatory for all South Koreans to take part in the national health insurance system. The NHIS is also responsible for maintaining national health examination programmes involving data from general health examinations of all insured employees, self-employed individuals and medical aid beneficiaries aged over 40 years; it is recommended for them to perform the health examination at least every 2 years. The health examination involves collection of information on body composition, blood profiles, blood pressure, self-reported lifestyles, self-reported physician-diagnosed disease and self-reported family history of disease.

The NHIS-HEALS cohort includes a wide variety of information collected between 2002 and 2015: health examination data and demographic and eligibility data (eg, inpatient and outpatient hospital records, medical bill, health insurance and medical aid beneficiaries). In the present analysis, we used health examination data collected between 2002 and 2008 to define the exercise frequency and all confounders. There was a change in the type of self-report methods in 2009; hence, health examination data collected in or after 2009 were not considered in the analysis due to the inability to harmonise variables. However, we used full follow-up data accrued from 2002 until 2015.

### Exposure

The primary exposure variable of this study was exercise frequency, assessed using questionnaires administered during the health examinations. The specific question asked was 'How many times per week do you engage in exercise that causes sweating?' Participants were asked to choose only one of the following 5 possible answers: none, 1–2 times/week, 3–4 times/week, 5–6 times/week and almost every day.

### Outcomes

In the present study, we evaluated 14 incident disease outcomes, namely, myocardial infarction; stroke, hypertension; type 2 diabetes mellitus; and stomach, colon, rectum, lung, liver, head and neck, pancreatic, kidney, gall bladder and oesophagus cancers. Participants' inpatient and outpatient hospital records (ie, non-fatal status) and death records (ie, fatal status) obtained through linkage with Statistics Korea were both classified according to the International Classification of Disease (ICD)-10 codes to classify different incidence types (online supplementary table 1). Additionally, blood pressure (eg, systolic ≥140 mm Hg, diastolic ≥90 mm Hg) and fasting glucose levels (eg, ≥126 mg/dL), both of which were measured during physical examinations, were used in conjunction with physicians' diagnosis information and ICD-10 codes to define incident hypertension and type 2 diabetes, respectively. Each incident disease outcome was defined as the first occurrence of either non-fatal or fatal respective disease cases. Incident disease cases were adjudicated using hospital and death records collected through 31 December 2015. The median follow-up was 13.0 years (IQR 10.2–11.3 years).

### Other covariates

The following covariates were included as confounders in the analyses: sex, body mass index (weight in kilograms

(kg) divided by height in metres squared ($m^2$)), systolic blood pressure, fasting glucose, total cholesterol, family history of heart disease, stroke or hypertension (only in models for incident myocardial infarction, stroke or hypertension), family history of diabetes (only in models for type 2 diabetes), family history of cancer (only in models for incident cancer outcomes), smoking status (never, previously, currently) and alcohol consumption (never, 2–3 times/month, 1–2 times/week, ≥3 times/week).

## Statistical analysis

Analyses were performed to summarise descriptive statistics (eg, means, SD, frequency and proportions) of each covariate and incident disease outcome for all participants and by exercise frequency category. Cox regression with age as the underlying timescale was used to estimate the associations of exercise frequency with each incident disease outcome, with adjustment for all the above-mentioned confounders as well as without any adjustment. HRs along with corresponding 95% CIs were calculated to evaluate relative risk of each incident disease outcome. Data were structured to enable the inclusion of exercise frequency and all confounders from both baseline and up to six repeated measures as time-updated covariates. This approach takes into account changes in exercise frequency as well as each confounder over time in relation to disease incidence. Individuals who reported no exercise served as a reference group for all comparisons. Effect modification by body mass index (<25, ≥25 $kg^2/m$), smoking status, alcohol consumption, family history of disease and sex was also examined based on Wald tests

of interaction terms in the fully adjusted models for each incident disease outcome. Visual inspections of log–log plots provided support for the assumptions of proportional hazards for all covariates. A sensitivity analysis where incident disease cases occurring during the first 2 years of follow-up were removed was performed to address reverse causality. Analyses were performed in Stata/SE V.14 (StataCorp LP, College Station, Texas, USA).

## Patient and public involvement

Neither patients nor members of the public were involved in this study.

## RESULTS

Of an initial sample of 512 190 individuals, 74 931 had missing data on at least one of the model covariates, and 179 405 had self-reported physician-diagnosed heart attack, stroke, hypertension (additionally, systolic ≥140 mm Hg or diastolic ≥90 mm Hg), diabetes (additionally, fasting glucose levels ≥126 mg/dL) or cancer at baseline, respectively. Excluding these individuals resulted in a final sample for analysis of 257 854 individuals (figure 1).

Individuals provided up to seven measures of exercise frequency and each confounder (ie, baseline plus six repeated measures). Participants' characteristics at baseline are summarised in table 1. Online supplementary table 2 summarises participants' characteristics at each repeat assessment. Individuals in the categories of 1–2, 3–4 or 5–6 times/week of exercise were slightly younger, but showed higher proportions of family history of disease

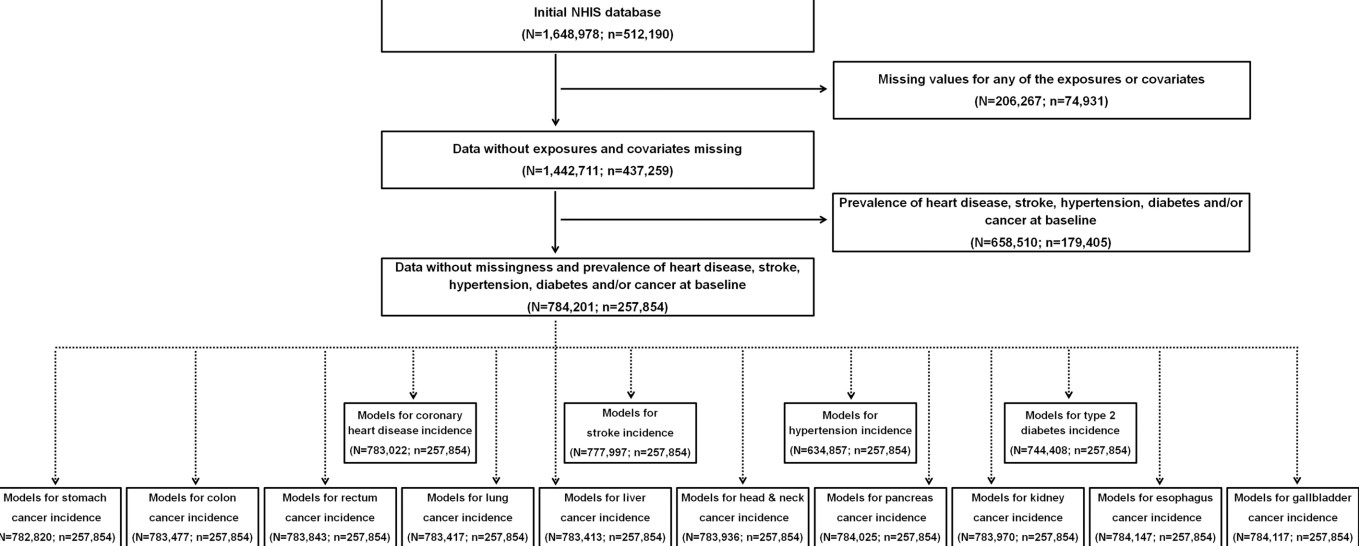

**Figure 1** A flow diagram. Note: 'N' indicates numbers of total observations (ie, participants who provided repeated measures are treated as separate observations) and 'n' indicates numbers of unique participants at baseline. Data without missingness and prevalence of major diseases were used to create final analysis datasets for different incident disease outcomes; while the number of unique participants at baseline is the same for all incident disease outcomes, the total number of observations varied due to the nature of time-updated covariate analyses (ie, censoring of subsequent time-updated covariates when an incident disease case occurs before the end date of repeated measures). Data were obtained from a prospective cohort, which has been established by linking physical examination data of over half a million South Korean adults (2002–2015) with the entire South Korean population's health insurance system. NHIS, National Health Insurance Service.

**Table 1**  Characteristics of the participants at baseline

| Variables | All (n=257 854) | Exercise frequency | | | | |
| --- | --- | --- | --- | --- | --- | --- |
| | | None (n=148 284) | 1–2 times/week (n=62 923) | 3–4 times/week (n=24 836) | 5–6 times/week (n=6676) | Almost everyday (n=15 135) |
| Sex, % | | | | | | |
| Men (%) | 50.5 | 43.8 | 63.3 | 57.5 | 52.9 | 49.6 |
| Women (%) | 49.5 | 56.2 | 36.7 | 42.5 | 47.1 | 50.4 |
| Age, years | 50.7 (8.7) | 51.5 (9.2) | 49.0 (7.5) | 49.3 (7.5) | 49.9 (7.8) | 53.5 (9.3) |
| Body mass index, kg$^2$/m | 23.5 (2.8) | 23.3 (2.9) | 23.6 (2.7) | 23.7 (2.6) | 23.7 (2.6) | 23.7 (2.7) |
| Systolic blood pressure, mm Hg | 116.8 (11.2) | 116.6 (11.3) | 117.1 (11.0) | 116.8 (11.1) | 116.8 (11.2) | 117.3 (11.3) |
| Diastolic blood pressure, mm Hg | 73.4 (7.9) | 73.1 (8.0) | 73.8 (7.8) | 73.5 (7.9) | 73.4 (8.0) | 73.4 (7.9) |
| Fasting glucose levels, mg/dL | 90.2 (12.3) | 90.1 (12.4) | 90.4 (12.2) | 89.9 (11.9) | 90.2 (12.3) | 90.2 (12.4) |
| Total cholesterol, mg/dL | 197.0 (36.7) | 196.5 (37.1) | 197.7 (36.2) | 197.8 (35.9) | 197.8 (36.1) | 197.4 (36.8) |
| Family history of heart disease, stroke or hypertension, % | 12.2 | 10.8 | 13.9 | 15.9 | 16.5 | 10.9 |
| Family history of cancer, % | 14.2 | 13.1 | 15.3 | 17.2 | 16.6 | 14.5 |
| Family history of diabetes, % | 6.3 | 5.4 | 7.2 | 8.5 | 9.2 | 5.7 |
| Smoking status, % | | | | | | |
| Never (%) | 68.1 | 71.9 | 59.1 | 65.2 | 68.8 | 71.7 |
| Previously (%) | 8.4 | 5.8 | 12.0 | 12.7 | 12.5 | 8.8 |
| Currently (%) | 23.6 | 22.2 | 28.9 | 22.1 | 18.8 | 19.5 |
| Alcohol consumption, % | | | | | | |
| Never (%) | 58.4 | 64.5 | 47.6 | 50.5 | 52.2 | 59.3 |
| 2–3 times/month (%) | 16.4 | 13.8 | 21.3 | 19.7 | 18.8 | 14.1 |
| 1–2 times/week (%) | 15.8 | 12.4 | 22.0 | 20.2 | 18.1 | 14.5 |
| ≥3 times/week (%) | 9.5 | 9.3 | 9.1 | 9.5 | 10.8 | 12.1 |
| Incident myocardial infarction, n (%) | 3047 (1.2) | 1741 (1.2) | 723 (1.1) | 276 (1.1) | 88 (1.3) | 219 (1.4) |
| Incident stroke, n (%) | 16 134 (6.3) | 9689 (6.5) | 3333 (5.3) | 1482 (6.0) | 390 (5.8) | 1240 (8.2) |
| Incident hypertension, n (%) | 120 203 (46.6) | 65 964 (44.5) | 30 623 (48.7) | 12 617 (50.8) | 3294 (49.3) | 7705 (50.9) |
| Incident type 2 diabetes, n (%) | 50 459 (19.6) | 27 128 (18.3) | 10 666 (6.5) | 5421 (21.8) | 1399 (21.0) | 3285 (21.7) |
| Incident stomach cancer, n (%) | 4788 (1.9) | 2672 (1.8) | 13 226 (21.0) | 489 (2.0) | 139 (2.1) | 328 (2.2) |
| Incident colon cancer, n (%) | 2711 (1.1) | 1424 (1.0) | 1160 (1.8) | 314 (1.3) | 90 (1.3) | 191 (1.3) |
| Incident rectum cancer, n (%) | 1494 (0.6) | 809 (0.6) | 692 (1.1) | 154 (0.6) | 46 (0.6) | 107 (0.7) |
| Incident lung cancer, n (%) | 3601 (1.4) | 2138 (1.4) | 796 (1.3) | 307 (1.2) | 85 (1.3) | 275 (1.8) |
| Incident liver cancer, n (%) | 2620 (1.0) | 1423 (1.0) | 680 (1.1) | 263 (1.1) | 75 (1.1) | 179 (1.2) |
| Incident pancreas cancer, n (%) | 864 (0.3) | 483 (0.3) | 205 (0.3) | 92 (0.4) | 24 (0.4) | 60 (0.4) |

**Table 1** Continued

| Variables | All (n=257854) | Exercise frequency | | | | |
| | | None (n=148284) | 1–2 times/week (n=62923) | 3–4 times/week (n=24836) | 5–6 times/week (n=6676) | Almost everyday (n=15135) |
| --- | --- | --- | --- | --- | --- | --- |
| Incident head and neck cancer, n (%) | 656 (0.3) | 377 (0.3) | 144 (0.2) | 73 (0.3) | 15 (0.2) | 47 (0.3) |
| Incident kidney cancer, n (%) | 589 (0.2) | 301 (0.2) | 153 (0.2) | 75 (0.3) | 16 (0.2) | 44 (0.3) |
| Incident gallbladder cancer, n (%) | 400 (0.2) | 219 (0.1) | 83 (0.1) | 43 (0.2) | 16 (0.2) | 39 (0.3) |
| Incident oesophagus cancer, n (%) | 352 (0.1) | 214 (0.1) | 75 (0.1) | 29 (0.1) | 6 (0.09) | 28 (0.2) |
| Median follow-up period, years (IQR) | 13.0 (12.2, 13.3) | 13.0 (12.2, 13.3) | 13.0 (12.2, 13.3) | 13.0 (12.2, 13.3) | 12.7 (12.2, 13.3) | 12.6 (12.2, 13.3) |

Values presented are means unless indicated as an 'n'. Values in parentheses are SD unless otherwise indicated. Data were obtained from a prospective cohort, which has been established by linking physical examination data of over half a million South Korean adults (2002–2015) with the entire South Korean population's health insurance system.

and lower proportions of never smoking or drinking alcohol, compared with those in the categories of none or almost every day of exercise. Across the seven time points (online supplementary figure 1), the proportion of individuals who reported no exercise decreased while the proportion who reported 1–2 or 3–4 times/week of exercise increased; there were no noticeable changes for the categories of 5–6 times/week or almost every day of exercise.

Overall, J-shaped associations were found between exercise frequency and incident myocardial infarction, stroke, hypertension and type 2 diabetes. HRs for these diseases were lowest in the middle categories of exercise frequency (eg, 3–4 or 5–6 times/week) (figure 2). There were no associations for the most frequent exercise category (eg, almost every day) with the incidence of myocardial infarction, stroke and type 2 diabetes.

J-shaped associations were also found for incident stomach, lung, liver and head and neck cancers (figure 3). Higher exercise frequencies (eg, 1–2, 3–4 times/week and almost every day) were associated with lower hazards of incident stomach cancer. No statistical significance was observed for incident colon, rectum, pancreas, kidney, gallbladder and oesophagus cancers. Crude event rates per 100000 person years in the middle categories of exercise frequency were relatively lower for incident rectum, and oesophagus cancers, but higher for incident pancreas, kidney and gallbladder cancers. Cox regression models with no adjustment for confounders (online supplementary figure 2) and a sensitivity analysis (online supplementary figure 3) in which incident cases occurring in the first 2 years of follow-up were removed both revealed nearly identical patterns of associations as the main analyses.

Figure 4 shows comparisons of results that showed statistical significance for multiplicative interaction terms between exercise frequency and each incident disease outcome. Strong J-shaped associations for incident hypertension were identified at each level of body mass index. J-shaped associations of exercise frequency with incident hypertension were strong only in the more favourable levels of smoking (eg, never, previously) and alcohol consumption (eg, never, 2–3 times/month, 1–2 times/week); no or weak associations were identified in the most harmful level of smoking (eg, current smokers) and alcohol consumption (eg, ≥3 times/week). J-shaped associations were evident at all levels of family history of CVD and sex for incident hypertension, and sex for incident type 2 diabetes. Exercise frequency was associated with incident lung cancer in non-obese individuals, but there was no evidence of association in obese individuals. All comparisons stratified by each potential effect modifier are presented in online supplementary figures 4 and 5.

## DISCUSSION

This is the first investigation examining the prospective associations of exercise with various incident non-communicable disease outcomes using multiple repeated measures of covariates in East Asian populations. We identified J-shaped associations of sweat-inducing exercise with incident myocardial infarction, stroke, hypertension, type 2 diabetes, stomach, lung, liver and head and neck cancers, with the greatest benefits being observed in the middle categories of exercise frequency (eg, 3–4 or 5–6 times/week): 1–2 times/week for head and neck cancer. These findings provide two important clinical and public health implications. First, prevention and management of non-communicable diseases in East Asians may benefit considerably from employing an exercise promotion approach in the context of combined non-communicable disease prevention. Mechanism research indicates that CVD and type 2 diabetes have similar biological pathways relating to exercise,[22 23] so an integrated prevention approach can be applied to control and manage

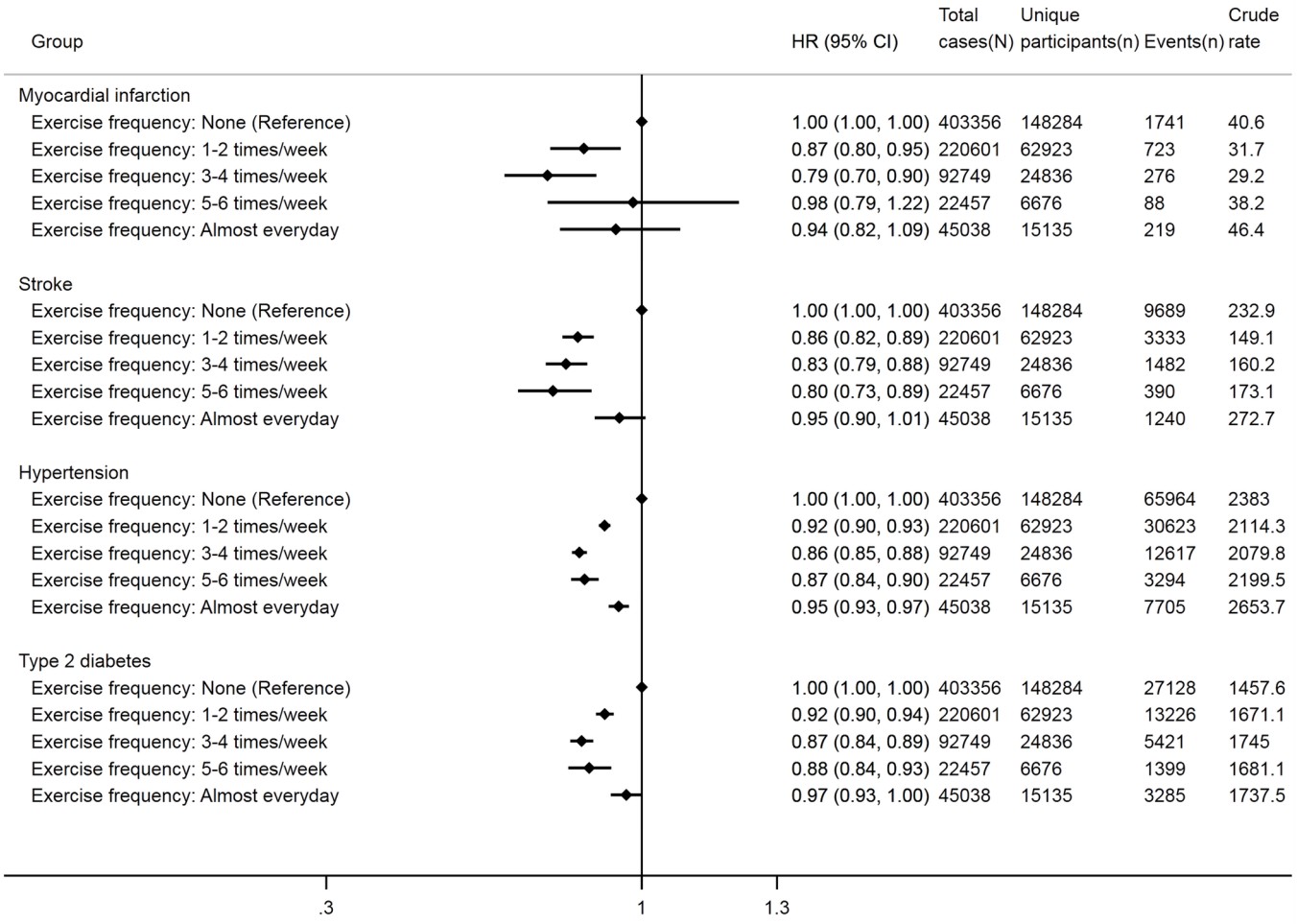

**Figure 2** Associations of exercise frequency with various incident cardiovascular disease outcomes. Cox regression models with age as the underlying timescale were adjusted for sex, body mass index, systolic blood pressure, fasting glucose levels, total cholesterol levels, family history of heart disease, stroke or hypertension (in models for myocardial infarction, stroke and hypertension) or diabetes (in models for type 2 diabetes), smoking status and alcohol consumption. Crude rates are per 100 000 person years. 'N' indicates numbers of total observations (ie, participants who provided repeated measures are treated as separate observations) and 'n' indicates numbers of unique participants at baseline. Data were obtained from a prospective cohort, which has been established by linking physical examination data of over half a million Korean adults (2002–2015) with the entire South Korean population's health insurance system.

these two diseases at a minimum.[5] Moreover, regular participation in exercise can induce favourable changes in intermediate cardiometabolic risk markers,[24] which are important predictors of typical non-communicable diseases. Hence, promoting exercise has great potential to act as an integrative behavioural strategy for preventing and controlling various non-communicable diseases simultaneously in East Asian populations.

Second, individuals who engage in exercise 3–4 or 5–6 times/week, rather than every day, may be able to reduce their risk of developing myocardial infarction, stroke, hypertension, type 2 diabetes, stomach, lung and liver cancers: 1–2 times/week for head and neck cancer. Similar J-shaped associations between high intensity exercise (eg, running) and CVD risk have also been reported in previous cohort studies of Western[25 26] and

Japanese adults.[27] Nevertheless, the present study as well as previous research[25–27] found that the risk of developing cardiovascular events in individuals who had the highest level of exercise was not noticeably higher compared with those who had the lowest level of exercise. No previous research in East Asians has found such J-shaped relationships between exercise or physical activity and other incident disease outcomes such as hypertension,[28–32] diabetes[33–40] and different type of cancers.[41–47] However, previous meta-analyses of cohort studies comprising predominantly Westerners found leisure-time physical activity to have curvilinear (but not J-shaped) associations with the incidence of type 2 diabetes,[48] and linear associations with the incidence of hypertension[49] and various site-specific cancers (liver, lung, head and neck, kidney, colon, rectal, bladder, gastric cardia, breast, endometrial,

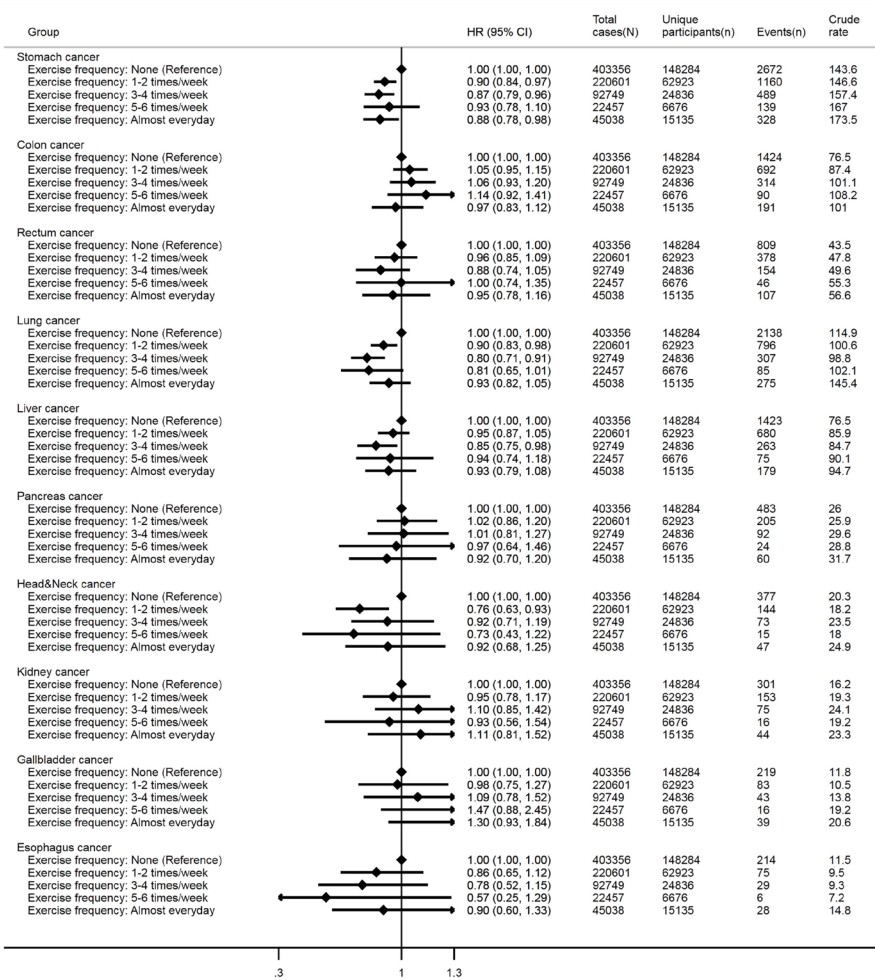

**Figure 3** Associations of exercise frequency with various incident cancer outcomes. Cox regression models with age as the underlying timescale were adjusted for sex, body mass index, systolic blood pressure, fasting glucose levels, total cholesterol levels, family history of cancer, smoking status and alcohol consumption. Crude rates are per 100 000 person years. 'N' indicates numbers of total observations (ie, participants who provided repeated measures are treated as separate observations) and 'n' indicates numbers of unique participants at baseline. Data were obtained from a prospective cohort, which has been established by linking physical examination data of over half a million South Korean adults (2002–2015) with the entire South Korean population's health insurance system.

myeloid leukaemia, myeloma, oesophageal adenocarcinoma).[50] While additional research is needed to confirm the J-shaped associations of exercise with various incident diseases in other samples of East Asians, findings of this research provide a strong rationale for the development and implementation of public health policies and clinical trials aimed at promoting a moderate level of sweat-causing exercise to minimise the risk of myocardial infarction, stroke, hypertension, type 2 diabetes, stomach, lung, liver and head and neck cancers.

Another finding of this research is that associations of sweat-inducing exercise with hypertension were modified by body mass index, smoking, alcohol consumption, family history of CVD and sex: lung cancer by body mass index and type 2 diabetes by sex. Notably, exercise frequency was not associated with hypertension in individuals who are smokers or drinking alcohol ≥3 times/week (except for 3–4 times/week of exercise). This observation provides some evidence that the harmful impacts of

smoking or binge drinking on hypertension[51–53] may not be offset completely by exercise. This, in turn, appears to advocate for the need for implementing a combined hypertension prevention strategy targeting promotion of exercise in conjunction with smoking cessation and reductions in alcohol consumption in East Asians.[13] For lung cancer, the null associations in individuals with body mass index ≥25 may be indicative of potential residual confounding through reported bias in smoking behaviours. Nonetheless, there was little evidence for effect modification for other disease comparisons, highlighting the importance of promoting exercise for the prevention of various non-communicable diseases in individuals at different levels of body mass index, smoking, alcohol consumption, family history of disease and sex.

This study has several notable strengths. First, we used data from a large prospective cohort study in which exercise and other risk markers were assessed on multiple occasions (up to seven times). Nearly 84% and 5% of the

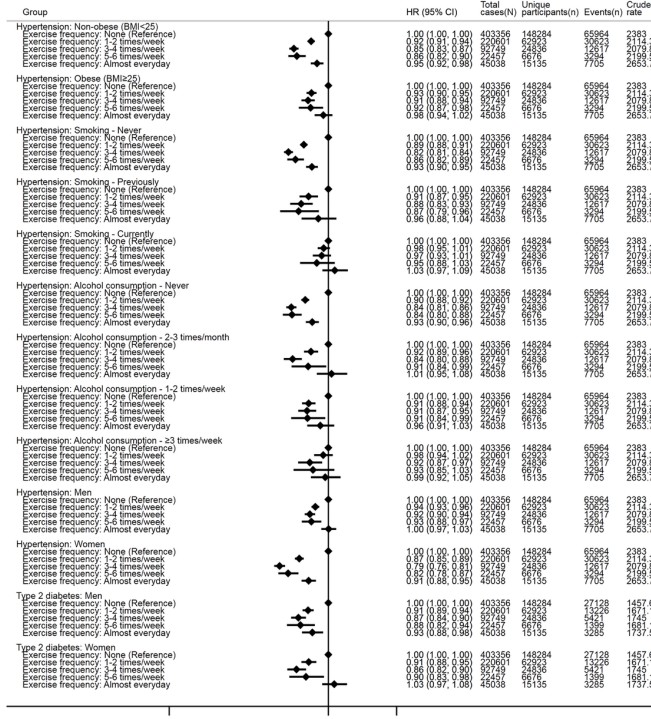

**Figure 4** Results from assessment of effect modification of sex, body mass index (BMI), smoking, alcohol consumption and family history of disease in the associations between exercise frequency and each incident disease outcome. Only associations for which the multiplicative interaction terms were statistically significant are presented. Cox regression models with age as the underlying timescale were adjusted for sex (not in models for effect modification by sex), BMI (not in models for effect modification by BMI), systolic blood pressure, fasting glucose levels, total cholesterol levels, family history (not in models for effect modification by family history of respective disease) of heart disease/stroke/hypertension (in models for myocardial infarction, stroke and hypertension), diabetes (in models for type 2 diabetes) or cancer (in models for each cancer), smoking status (not in models for effect modification by smoking status) and alcohol consumption (not in models for effect modification by alcohol consumption). Crude rates are per 100 000 person years. 'N' indicates numbers of total observations (ie, participants who provided repeated measures are treated as separate observations) and 'n' indicates numbers of unique participants at baseline. P values for multiplicative interactions with exercise frequency are as follows; outcome—hypertension: BMI (0.049), smoking (<0.001), alcohol consumption (<0.001), family history of CVD (0.029) and sex (<0.001); outcome—lung cancer: BMI (0.016), and outcome—type 2 diabetes: sex (0.012). Data were obtained from a prospective cohort, which has been established by linking physical examination data of over half a million South Korean adults (2002–2015) with the entire South Korean population's health insurance system.

full participants provided one and six repeated measures of all covariates, respectively. Compelling evidence indicates that the risk of regression dilution can be reduced using repeated measures of exposure and confounders.[14] Moreover, we examined the dose–response relationship

of exercise frequency with a wide variety of specific types of incident non-communicable disease outcomes simultaneously using inpatient and outpatient diagnosis data as well as mortality data. The large sample size (n=257854) is another strength.

This study has some limitations. Findings of this study may not be generalisable to adult populations of other ethnic origins. Due to the observational nature of this research, no strong causal inference can be drawn about the exercise-incident disease relationships. In addition, the accuracy of hospital admission records is uncertain, although the accuracy of death records from Statistics Korea was found to be 92% in previous research.[53] No information about medication use was available in the cohort data, so we could not use it as a potential confounder and another condition when defining disease status (eg, hypertension, type 2 diabetes) at both baseline and follow-up. Furthermore, no exercise duration was assessed; hence, inference was made purely based on exercise frequency. Moreover, ICD-10 codes for sex-specific cancers (eg, prostate and breast cancers) were masked due to the data management policy set forth by the NHIS, so it was not possible to examine such cancers in the present study. The lack of data on diet, which is another behavioural risk marker for non-communicable diseases,[12] is another limitation. Moreover, a sizeable proportion (n=74931; 14.6%) of individuals were excluded due to the missing information on the covariates. Another limitation is that the measurement methods to assess the covariates were not standardised across the different medical institutes participating in the NHIS-HEALS cohort.

## CONCLUSION

Individuals who engaged in sweat-inducing exercise around 3–6 times/week (as opposed to every day) generally had the lowest risk of developing myocardial infarction, stroke, hypertension, type 2 diabetes, stomach, lung, liver and head and neck cancers. These findings were generally applicable to different subpopulations as stratified by body mass index, smoking, alcohol consumption, family history of disease and sex. Public health and lifestyle interventions should promote a moderate level of sweat-inducing exercise as a behavioural strategy for prevention and control of non-communicable diseases in a wider population of East Asians.

**Contributors** YK designed this study, performed statistical analysis and drafted an initial version of the manuscript. SS, S-mH and SHJ all contributed to conceptualising the study idea and developing the analytical plans, and provided assistance with statistical analysis. All authors critically reviewed, approved the final version of the manuscript and agreed to be responsible for all facets of this work.

**Funding** This work was supported by the National R&D Program for Cancer Control, Ministry of Health & Welfare, Republic of Korea (Grant 1631020 to SHJ), The Korean Health Technology R&D Project, Ministry of Health & Welfare, Republic of Korea (Grant HI14C2686 to SHJ) and the Medical Research Council (Grant MC_UU_12015/1 to SS). The funders had no role in study design, data collection and analysis, decision to publish or preparation of the manuscript.

**Competing interests** None declared.

**Patient consent for publication** Not required.

**Ethics approval** This research was approved by the Institutional Review Board (4-2017-0051) of the Yonsei University's Severance Hospital in Republic of Korea.

**Provenance and peer review** Not commissioned; externally peer reviewed.

**Data sharing statement** Data sharing is not applicable because no informed consent for data sharing was obtained from the participants.

**Open access** This is an open access article distributed in accordance with the Creative Commons Attribution 4.0 Unported (CC BY 4.0) license, which permits others to copy, redistribute, remix, transform and build upon this work for any purpose, provided the original work is properly cited, a link to the licence is given, and indication of whether changes were made. See: https://creativecommons.org/licenses/by/4.0/.

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
