## [Reviewer comments · BMJ Open]

ARTICLE DETAILS

TITLE (PROVISIONAL)	Exercise and incidence of myocardial infarction, stroke, hypertension, type 2 diabetes and site-specific cancers: A prospective cohort study of 257,854 adults in South Korea.
AUTHORS	Kim, Youngwon; Sharp, Stephen; Hwang, Se-mi; Jee, Sun Ha

VERSION 1 – REVIEW

REVIEWER	Reviewer name: Nole C Barengo Institution and Country: Department of Medical and Population Health Sciences Research, Herbert Wertheim College of Medicine, Florida International University. Miami, USA. Competing interests: None declared
REVIEW RETURNED	06-Aug-2018

GENERAL COMMENTS	This present work assessed the associations of exercise frequency with the incidence of myocardial infarction, stroke, hypertension, type 2 diabetes and 10 different cancer outcomes in a large South-Korean population cohort. The study is methodologically well conducted and adds valuable information in physical activity research in a different population than European or American. However, I have some suggestions on how to improve the manuscript before possible publication. Specific comments Abstract • Please, remove the first sentence from the abstract (“Little is known...”) and start directly presenting the objective (“The objective of this study...”).• Add the time frame to settings (during xxxx and xxxx) Introduction • Reference 10 (Diabetes rates...), clarify if this is diabetes incidence or a prevalence (prevalence is not a rate). In case of diabetes, this may be due to an increase in screening activities; similarly define whether you mean CVD prevalence, incidence rate or cumulative incidence (reference 3). Methods • Please, add more information in regard the sampling methodology. It was mentioned that the random sample used stratification by sex, age, employment status and income. Did it also include other factors such as weighting or clustering? Was stratification by age done by age-groups (if yes, what were the age groups, employment status groups and income groups)?• Confounders: Please, replace the word confounder by “covariates” as it is not certain whether they are all confounders or covariates/other variables.
--

	 • Please, provide more detailed information on these covariates. Methods of measuring fasting glucose, cholesterol, etc., definition of BMI, BMI categories (usually different from European populations), how was weight and height measured, how was blood pressure measured/recorded, etc. • Statistical analysis: how did you check whether the covariates were confounders or not (stepwise regression manually or programmed?), as you mentioned effect modification, did you include the interaction terms in the adjusted models or how did you assess interaction? Please, mention that hazard ratios and 95% confidence intervals were calculated. Add information on the assessment of and presentation of the baseline characteristics (Table 1) at the beginning of the statistical analysis section. Discussion  • Did you perform any sensitivity analysis (best vs worst case scenario) as almost 40% of the people were excluded from the analysis due to missing information on some of the variables? According to that, do you suggest that the point estimates received is an over or under-estimation of the “true” effect? Please, discuss this in the limitations of the study section. Tables and figures  • The authors tried to combine both a figure and table into one. However, I am more interested in seeing the unadjusted and adjusted HR in each table instead of the figure element, specifically as the figure provided the same information as the numbers in tables (HR, 95% CI). • To draw conclusions whether there is effect modification, the results of the interaction terms of the adjusted models need to be presented. I believe that stratification alone does not provide sufficient evidence on interaction, but rather, controls for confounding. Alternatively, you may omit mentioning effect modification but discuss the different stratum-effects.
--	---

REVIEWER	Reviewer name: Elizabeth Dean Institution and Country: University of British Columbia, Canada Competing interests: None declared
REVIEW RETURNED	09-Aug-2018

GENERAL COMMENTS	Review of ‘Exercise and incidence of myocardial infarction, stroke, hypertension, type 2 diabetes and site-specific cancer’ Overview The objective of this study was ‘to examine longitudinal associations of exercise frequency with the incidence of myocardial infarction, stroke, hypertension, type 2 diabetes, and 10 different cancer outcomes.’ Based on a prospective cohort design, the investigators used ‘Physical examination data linked with the entire Korean population health insurance system’ as a basis for extracting the relevant data. Participants included 257,854 Korean adults with up to seven repeated measures or exercise and confounders. Primary outcome measures included each disease incidence based on both fatal and non-fatal health records. The investigators report that ‘Compared with no exercise category, the middle category of exercise frequency (≤ 6times/week) showed the lowest rate of (across disease diagnoses), exhibiting J-shaped associations.’ Further, they observed ‘..little evidence of effect modification by body mass index, smoking, alcohol consumption, family history of disease, and sex in these associations.’ In turn, they concluded that ‘Public
---

health and lifestyle interventions should promote moderate levels of exercise as a behavioral prevention strategy for non-communicable diseases in a wider population of East Asians.'

Substantive Comments

The premise of the work is interesting and the large sample from the South Korean data base is impressive. I would like to see the manuscript re-written to enhance its scientific and technical precision in addition to writing quality. I recommend rewriting the Ms. to better reflect the findings. For example, exercising 1-6 times a week at a level that causes sweating cannot be considered a 'middle category'. I believe the data need to be re-worked to get at the essence of the data. Perhaps, partitioned somehow. This will require some thought as the levels are currently: 1-2, 3-4, 5-6, and almost every day. Given the recommendation of moderately-intense or vigorous exercise for general fitness is 3-5 times a week at 70-85% of age predicated maximal heart rate, categories of exercise of <3 times a week, 3-5 times a week, and over 5 times a week, would have been more useful in some ways. The intensity parameter seems to have been overlooked by the investigators, however despite that, the fact the data that were collected apparently specified 'sufficient to make you sweat' makes the findings more useful. I appreciate that the investigators used an existing database with no control over the questions.

The investigators emphasize exercise frequency however they confound frequency with 'intensity'. The exercise literature states that of frequency, intensity, and duration, 'intensity' is the most important for aerobic fitness and conditioning benefit. Thus, each of frequency should really be paired with mention of sweat-inducing exercise.

Be clear about terminology re East Asians (who?) and Koreans. Life is very different between the north and the south which would be a major confounding factor. I suggest always making reference to South Koreans, unless North Koreans have been studied in other research.

Editorial

The manuscript needs to be copyedited thoroughly for composition and grammar, in addition to journal style and formatting including the references (in which there are innumerable errors).

Given the cohort studied and the conclusion, the title should make specific reference to South Koreans.

BMJ Open likely defaults to UK spelling. Check the instructions to authors.

In scientific writing, other than in tables, it is best to avoid short hand notation such as the use of & and < and >.

Page 7 Line 5 through 7. Delete 'etc.' at the end of the sequence, as it is redundant with the use of e.g. at the beginning (note, e.g. and the use of i.e. should end with commas as well).

Page 7 Line 43. Edit to the plural 'cancers' here and elsewhere when referring to multiple cancer diagnostic categories.

Page 7 Line 39 (sentence) should be something like:

In the present study, we evaluated 17 incident disease outcomes, namely, myocardial infarction; stroke, hypertension; type 2 diabetes mellitus; and stomach, colon, rectum, lung, liver, head and neck, pancreatic, kidney, gall bladder and esophagus cancers.

	Page 8 Line 7. 'censored' does not fit. Note that mmHg should be mm Hg (with a space) Several table/figure legends are jagged re justification; use left justification only unless otherwise instructed in the guidelines.
--	---

VERSION 1 – AUTHOR RESPONSE

Reviewer: 1

Reviewer Name: Nole C Barengo

Institution and Country: Department of Medical and Population Health Sciences Research
Herbert Wertheim College of Medicine, Florida International University
Miami, USA

Please state any competing interests or state 'None declared': None declared

Please leave your comments for the authors below This present work assessed the associations of exercise frequency with the incidence of myocardial infarction, stroke, hypertension, type 2 diabetes and 10 different cancer outcomes in a large South-Korean population cohort. The study is methodologically well conducted and adds valuable information in physical activity research in a different population than European or American. However, I have some suggestions on how to improve the manuscript before possible publication.

Specific comments

Abstract

- Please, remove the first sentence from the abstract (“Little is known...”) and start directly presenting the objective (“The objective of this study...”).
- Response: We removed the first sentence in the Abstract.
- Add the time frame to settings (during xxxx and xxxx)
- Response: We have added “from 2002 to 2015” to settings

Introduction

- Reference 10 (Diabetes rates...), clarify if this is diabetes incidence or a prevalence (prevalence is not a rate). In case of diabetes, this may be due to an increase in screening activities; similarly define whether you mean CVD prevalence, incidence rate or cumulative incidence (reference 3).
- Response: We have made a change to specifically indicate diabetes prevalence (Reference 10) and CVD prevalence (Reference 3).

Methods

- Please, add more information in regard the sampling methodology. It was mentioned that the random sample used stratification by sex, age, employment status and income. Did it also include other factors such as weighting or clustering? Was stratification by age done by age-groups (if yes, what were the age groups, employment status groups and income groups)?
- Response: We acknowledge that the methods section needed more in-depth descriptions about stratification. We have added the following statements to specifically indicate the different levels of sex, age, employment status, and income used in the stratifications strategy:

“(stratified by 2 groups of sex [males and females], 18 groups of age ranges [less than 1 year, 1-4 years, every 5 years between 5-79 years, and more than 80 years], 3 groups of employment status [insured employees, self-employed individuals and medical aid beneficiaries] and 41 groups of income levels [upper 20% for insured employees, lower 20% for insured self-employed individuals and the lowest level for medical aid beneficiaries]21)”

- Confounders: Please, replace the word confounder by “covariates” as it is not certain whether they are all confounders or covariates/other variables.

- Response: Thanks for the suggestion. We have changed Confounders to “Other covariates” to differentiate them from the main covariate variable (Exercise Frequency) which served as the main exposure variable in the analyses.

- Please, provide more detailed information on these covariates. Methods of measuring fasting glucose, cholesterol, etc., definition of BMI, BMI categories (usually different from European populations), how was weight and height measured, how was blood pressure measured/recorded, etc.

- Response: NHIS-HEALS was established based on data obtained as part of general health check-up programs implemented at different medical institutions in South Korea. While all the covariates were measured, the measurement methods used were not standardized across the different medical institutes. We acknowledge it as a limitation, so add it as another limitation in the Discussion section as follows:

“Another limitation is that the measurement methods to assess the covariates were not standardized across the different medical institutions participating in the NHIS-HEALS cohort.”

- We used a BMI value of 25kg/m² when evaluating the effect modification by BMI. We have indicated it in the Statistical Analyses section.

- Statistical analysis: how did you check whether the covariates were confounders or not (stepwise regression manually or programmed?), as you mentioned effect modification, did you include the interaction terms in the adjusted models or how did you assess interaction? Please, mention that hazard ratios and 95% confidence intervals were calculated. Add information on the assessment of and presentation of the baseline characteristics (Table 1) at the beginning of the statistical analysis section.

- Response:

- The decision to include the confounders was made ‘a-priori’ based on the established body of literature showing the potential confounding roles of these variables in the exercise-disease associations. We did not use stepwise regression to determine the list of confounders.

- We acknowledge that there was not enough information about the tests for effect modification. So we have provided more detailed descriptions about how we evaluated effect modification in the Statistical analysis section as follows:

“Effect modification by body mass index (<25, ≥25kg²/m), smoking status, alcohol consumption, family history of disease and sex was also examined based on Wald tests of interaction terms in the fully adjusted models for each incident disease outcome”

The specific p-values from the Wald tests of interaction terms are provided in the Supplementary Figures 4 and 5.

- We have provided a statement indicating the calculation of hazard ratios and 95% confidence intervals as follows:

“Hazard ratios along with corresponding 95% confidence intervals were calculated to evaluate relative risk of each incident disease outcome.”

- We have provided a statement at the beginning of the Statistical analysis section describing the analyses of descriptive statistics as follows:

“Analyses were performed to summarise descriptive statistics (e.g., means, standard deviations, frequency, and proportions) of each covariate and incident disease outcome for all participants and by exercise frequency category.”

Discussion

- Did you perform any sensitivity analysis (best vs worst case scenario) as almost 40% of the people were excluded from the analysis due to missing information on some of the variables? According to that, do you suggest that the point estimates received is an over or under-estimation of the “true” effect? Please, discuss this in the limitations of the study section.
- The number of individuals that were excluded due to the missing information on the key covariates is 74,931, which is about 15% of the initial sample of individuals (n=512,190). It is not clear whether the exclusion of these individuals led to an over or under-estimation of the true effects, but we agree that it is a substantial number of individuals, so we have indicated it as one of the limitations of the study as follows:
“Moreover, a sizeable proportion (n=74,931; 14.6%) of individuals were excluded due to the missing information on the covariates.”

Tables and figures

- The authors tried to combine both a figure and table into one. However, I am more interested in seeing the unadjusted and adjusted HR in each table instead of the figure element, specifically as the figure provided the same information as the numbers in tables (HR, 95% CI).
- Response: We believe that the figures show some of the essential results in a very clear manner. The forest plot on its own (without the specific numbers (HR, 95% CI)) would not show the specific HR values and 95% CIs for each comparison group. So we decided not to separate the HR and CI values from the forest plot.
- We provided results from unadjusted models (i.e. models not adjusted for any confounders) in Supplementary Figure 2.
- To draw conclusions whether there is effect modification, the results of the interaction terms of the adjusted models need to be presented. I believe that stratification alone does not provide sufficient evidence on interaction, but rather, controls for confounding. Alternatively, you may omit mentioning effect modification but discuss the different stratum-effects.
- Response: As we mentioned above, we provided the specific p-values of interaction terms in the Supplementary Figures 4 and 5.

Reviewer: 2

Reviewer Name: Elizabeth Dean

Institution and Country: University of British Columbia
Canada

Please state any competing interests or state ‘None declared’: None declared

Please leave your comments for the authors below Review of ‘Exercise and incidence of myocardial infarction, stroke, hypertension, type 2 diabetes and site-specific cancer’

Overview

The objective of this study was ‘to examine longitudinal associations of exercise frequency with the incidence of myocardial infarction, stroke, hypertension, type 2 diabetes, and 10 different cancer outcomes.’ Based on a prospective cohort design, the investigators used ‘Physical examination data linked with the entire Korean population health insurance system’ as a basis for extracting the relevant data.

Participants included 257,854 Korean adults with up to seven repeated measures on exercise and confounders. Primary outcome measures included each disease incidence based on both fatal and non-fatal health records. The investigators report that 'Compared with no exercise category, the middle category of exercise frequency (≤ 6 times/week) showed the lowest rate of (across disease diagnoses), exhibiting J-shaped associations.' Further, they observed '...little evidence of effect modification by body mass index, smoking, alcohol consumption, family history of disease, and sex in these associations.' In turn, they concluded that 'Public health and lifestyle interventions should promote moderate levels of exercise as a behavioral prevention strategy for non-communicable diseases in a wider population of East Asians.'

Substantive Comments

The premise of the work is interesting and the large sample from the South Korean data base is impressive. I would like to see the manuscript re-written to enhance its scientific and technical precision in addition to writing quality. I recommend rewriting the Ms. to better reflect the findings. For example, exercising 1-6 times a week at a level that causes sweating cannot be considered a 'middle category'. I believe the data need to be re-worked to get at the essence of the data. Perhaps, partitioned somehow. This will require some thought as the levels are currently: 1-2, 3-4, 5-6, and almost every day. Given the recommendation of moderately-intense or vigorous exercise for general fitness is 3-5 times a week at 70-85% of age predicated maximal heart rate, categories of exercise of <3 times a week, 3-5 times a week, and over 5 times a week, would have been more useful in some ways. The intensity parameter seems to have been overlooked by the investigators, however despite that, the fact the data that were collected apparently specified 'sufficient to make you sweat' makes the findings more useful. I appreciate that the investigators used an existing database with no control over the questions.

- Response: Thanks for the suggestions.
- One of the recommendations was reclassifying the current 5 categories of exercise frequency to define 3 categories of exercise (i.e. <3 times, 3-5 times and >5 times/week). This suggested method is essentially collapsing the 5 categories into 3 categories by combining the first 2 categories (i.e., no exercise and 1-2 times/week) and the last 2 categories (i.e. 5-6 times/week and almost everyday).
- However, we decided to maintain our current 5 categories of exercise to directly show the relative risk for each of the 5 exercise categories. There are many instances where large differences in relative risk were observed when changing from 'no exercise' to '1-2 times/week', and from '5-6 times/week' to 'almost everyday'. So, using only 3 combined categories may mask some of the important associations.
- As for the comment "exercising 1-6 times a week at a level that causes sweating cannot be considered a 'middle category'", we acknowledge that the relatively broader range of exercise frequency (i.e. 1-6 times/week) that we indicated as 'middle categories' can potentially misguide readers. In fact, the lowest risk was observed in the 3-4 or 5-6 times/week categories for all incident disease outcomes, only except for head & neck cancer for which 1-2 times/week had the lowest risk. So, we made this clearer in the Abstract, and main text.

The investigators emphasize exercise frequency however they confound frequency with 'intensity'. The exercise literature states that of frequency, intensity, and duration, 'intensity' is the most important for aerobic fitness and conditioning benefit. Thus, each of frequency should really be paired with mention of sweat-inducing exercise.

- Response: We agree that exercise intensity (i.e., sweat-inducing) was implicit in the definition of exercise frequency used in the original questionnaires so we added "sweat-inducing" or "sweat-causing" where appropriate throughout the manuscript.

Be clear about terminology re East Asians (who?) and Koreans. Life is very different between the north and the south which would be a major confounding factor. I suggest always making reference to South Koreans, unless North Koreans have been studied in other research.

- Response: We have now made it clear that we used a sample of South Koreans. No data from North Koreans were used in this research.

Editorial

The manuscript needs to be copyedited thoroughly for composition and grammar, in addition to journal style and formatting including the references (in which there are innumerable errors).

- Response: We made changes to improve the readability of the manuscript. We have also made changes on the reference list.

Given the cohort studied and the conclusion, the title should make specific reference to South Koreans.

- Response: We are not aware that mentioning a specific population investigated in the title is usual practice for general research articles. We have specifically indicated that we studied a cohort dataset of South Koreans in both the Abstract and the main text. Also, we have added the specific design of this study (i.e. a prospective cohort study) to the title, as suggested by the editor.

BMJ Open likely defaults to UK spelling. Check the instructions to authors.

- Response: We have now used UK spelling.

In scientific writing, other than in tables, it is best to avoid short hand notation such as the use of & and < and >.

- Response: We have changed the use of '>' to 'over' but maintained the use of '<' in a sentence (Line 174) where it helps to show the BMI categories in a concise manner: "Effect modification by body mass index (<25, ≥25kg2/m)...."
- We have maintained the use of '&' to emphasize the fact that we investigated head cancer and neck cancer altogether as a single cancer outcome (i.e., head & neck cancer). If we use 'and' instead of '&' in this context, readers may potentially think that we investigated head cancer and neck cancer as two separate cancer outcomes. For example, "risk of myocardial infarction, stroke, hypertension, type 2 diabetes, stomach, lung, liver and head & neck cancers" seems more straightforward to understand than "risk of myocardial infarction, stroke, hypertension, type 2 diabetes, stomach, lung, liver, head and neck cancers" or "risk of myocardial infarction, stroke, hypertension, type 2 diabetes, stomach, lung, liver and head and neck cancers"

Page 7 Line 5 through 7. Delete 'etc.' at the end of the sequence, as it is redundant with the use of e.g. at the beginning (note, e.g. and the use of i.e. should end with commas as well).

- Response: We have made these changes.

Page 7 Line 43. Edit to the plural 'cancers' here and elsewhere when referring to multiple cancer diagnostic categories.

- Response: We made this change where needed throughout the manuscript.

Page 7 Line 39 (sentence) should be something like:

In the present study, we evaluated 17 incident disease outcomes, namely, myocardial infarction; stroke, hypertension; type 2 diabetes mellitus; and stomach, colon, rectum, lung, liver, head and neck, pancreatic, kidney, gall bladder and esophagus cancers.

- Response: We have made the suggested change.

Page 8 Line 7. 'censored' does not fit.

- Response: Thanks for the suggestion. We have now changed it to read as follows: "Incident disease cases were adjudicated using hospital and death records collected through December 31st, 2015."

Note that mmHg should be mm Hg (with a space) Several table/figure legends are jagged re justification; use left justification only unless otherwise instructed in the guidelines.

- Response: We have changed mmHg to mm Hg. We would work with the journal's editing specialists if any further formatting changes need to be made.

FORMATTING AMENDMENTS (if any)

Required amendments will be listed here; please include these changes in your revised version:

- Kindly re-upload each figure under 'Image' file designation with at least 300 dpi resolution and at least 90mm x 90mm of width.

- Response: We have re-uploaded our figure files.

- Please re-upload your supplementary files in PDF format.

- Response: We have now uploaded our supplementary files in PDF.